# Comparative Study of Short-Term Efficacy and Safety of Radical Surgery with or without Hyperthermic Intraperitoneal Chemotherapy in Colorectal Cancer with T4 Stage: A Propensity Score Matching Analysis

**DOI:** 10.3390/jcm12031145

**Published:** 2023-02-01

**Authors:** Xikai Guo, Yao Lin, Chu Shen, Yuan Li, Xinyu Zeng, Jianbo Lv, Fan Xiang, Tuo Ruan, Chuanqing Wu, Kaixiong Tao

**Affiliations:** Department of Gastrointestinal Surgery, Union Hospital, Tongji Medical College, Huazhong University of Science and Technology, Wuhan 430022, China

**Keywords:** hyperthermic intraperitoneal chemotherapy, T4 colorectal cancer, peritoneal carcinomatosis, propensity score matching, efficacy, safety

## Abstract

Background: Hyperthermic intraperitoneal chemotherapy (HIPEC) in T4 colorectal cancer (CRC) remains controversial. The study aimed to explore the safety and efficacy of radical surgery (RS) with HIPEC in T4 CRC. Methods: Adverse events after HIPEC were estimated by Common Terminology Criteria for Adverse Events version 5.0. The efficacy was evaluated using recurrence-free survival (RFS) and overall survival (OS). Propensity score matching (PSM) was used to reduce the effects of confounders between groups. Results: Of the 417 patients (263 men and 154 women), 165 patients were treated with RS + HIPEC and 252 patients with RS alone. There was no significant difference in the incidence of all adverse events after PSM. Overall RFS and OS were not significantly different at 24 months (*p* = 0.580 and *p* = 0.072, respectively). However, in patients with T4b stage CRC (92.1% vs. 77.3%, *p* = 0.048) and tumor size ≥ 5 cm (93.0% vs. 80.9%, *p* = 0.029), RFS in the two groups showed a significant difference at 24 months. Conclusions: In summary, the safety of HIPEC in T4 CRC was confirmed. Compared with RS, though RS + HIPEC did not benefit the overall cohort at 24 months, RS + HIPEC could benefit patients with T4b stage CRC and tumor size ≥ 5 cm in RFS.

## 1. Introduction

Colorectal cancer (CRC) is one of the most common malignant tumors of the digestive system. The latest data show that in the United States, CRC had the third highest incidence in both males and females [1]. In Asian populations, a 2018 Chinese cancer statistics report showed that CRC incidence and mortality ranked third and fifth among all malignancies [2]. The common distant metastases of CRC include liver, lung, and peritoneal carcinomatosis (PC), among which PC has the worst prognosis and shortest survival [3,4]. The prognosis of CRC with PC is poor, and the median overall survival (OS) is only 5–7 months [5]. Studies have shown that the 3-year peritoneal metastasis rate of pT4 colorectal cancer is as high as 20–36.7% [6]. So, the T4 stage is a high-risk factor for PC in patients with CRC.

For T4 CRC, many reports have described positive intraperitoneal treatments, including radical surgery (RS) with hyperthermic intraperitoneal chemotherapy (HIPEC), to improve the quality of life and survival of patients with PC [7]. RS combined with HIPEC is currently the recommended treatment method for CRC with PC, according to the consensus in many countries [8,9,10]. However, whether HIPEC improves peritoneal metastasis-free survival in patients with T4 CRC remains controversial. Klaver et al. reported that the 18-month peritoneal metastasis-free survival of patients with T4 or perforated colon cancer did not improve with the adjuvant HIPEC plus oxaliplatin [11], while A. Arjona-Sanchez et al. reported that RS + HIEPC with mitomycin C for locally advanced colon cancer could improve the loco-regional control rate (97% vs. 87%, *p* = 0.025) [12].

There is a lack of convincing results elucidating the safety and efficacy of RS combined with HIPEC in patients with CRC. Therefore, this large-scale retrospective study aimed to explore the safety and efficacy of tumor cytoreductive surgery combined with HIPEC in patients with T4 CRC.

## 2. Patients and Methods

### 2.1. Study Cohort

We reviewed the hospital records of patients diagnosed with T4 CRC and treated with RS with or without HIPEC, from January 2019 to April 2020, at Wuhan Union Hospital, Tongji Medical College, Huazhong University of Science and Technology. Participants were included with the following criteria: (1) confirmation of colorectal adenocarcinoma by postoperative pathology (pT ≥ 4), (2) no distant metastasis or PC detected before the operation through computed tomography (CT) or magnetic resonance imaging (MRI), (3) adjuvant chemotherapy taken after discharge, and (4) a tumor location in patients with rectal cancer above the peritoneal reflection. Preoperative MRI and colonoscopy showed at least 7 cm from the lower border of the tumor to the anal verge (above the peritoneal reflection). The exclusion criteria included: (1) incomplete clinical and pathological data preoperatively, (2) detectable tumor metastasis found during the operation, (3) patients with a history of other malignancies within 5 years, and (4) emergency surgery due to perforation, ileus, etc. Institutional ethical approval [2021] LUNSHENGZI (0601) was received for the study.

The patients were classified into (a) the experimental group: RS + HIPEC, depending on whether they received HIPEC treatment, and (b) the control group: RS alone. This retrospective study was approved by the institutional review board, and patient information and privacy were closely protected.

### 2.2. Data Collection

Patient data were collected through the hospital’s database and included sex, age, height, weight, history, American Society of Anesthesiologists (ASA) classification, operation method, tumor location, maximum tumor diameter, tumor histological type, surgical margin, nerve invasion, vascular invasion, lymphatic metastasis, depth of invasion, length of postoperative hospital stay, and HIPEC-related information. Body mass index (BMI) and the Charlson comorbidity index (CCI) were used to evaluate height and weight and basic preoperative medical history, respectively. Additionally, lymph node metastasis and depth of invasion were evaluated using the American Joint Committee on Cancer TNM staging system.

The Common Terminology Criteria for Adverse Events version 5.0 was used to record adverse events of grade 2 and above that occurred postoperatively with or without HIPEC. The adverse events included anemia, hypoalbuminemia, myelosuppression, wound complications, abdominal infection, pulmonary infection, postoperative bleeding, gastroparesis, anastomotic leakage, ileus, electrolyte disorder (Appendix A).

### 2.3. Follow-Up

The latest time of follow-up was in June 2022. Patients were followed up every 3 months during the first 2 years after surgery. Follow-up data were collected from outpatient visits or re-examination results that met the follow-up requirements sent by patients through the network. Based on individual patient necessity, CT of lungs, abdomen, and pelvis, enhanced CT, MRI, or colonoscopy were performed during outpatient follow-up.

The treatment effectiveness was evaluated using: (1) treatment efficacy indicators, including tumor recurrence, metastasis, and death; (2) recurrence-free survival (RFS), which stood for the time from RS until the patient’s recurrence (local recurrence and peritoneal metastasis) or last contact, and the time point of progression was the date of the first detectable new lesions; (3) OS, which was the time from the initial RS to death due to any cause or the end of follow-up.

### 2.4. Treatment

None of the patients included in this study received neoadjuvant therapy or radiotherapy. All patients had been informed about the operation procedure and HIPEC therapy due to their cT4 preoperative staging and voluntarily signed an informed consent form for the procedure. HIPEC was conducted according to the formal standards and specifications of the clinical application of HIPEC in China, constituted by the Peritoneal Surface Oncology Committee of the China Anti-Cancer Association [13]. Clinicians performed radical resection of CRC (resection of the corresponding colon and rectum plus regional lymph node dissection), following the principles of mesorectal excision and tumor-free operation, and the surgical procedure was referred to the Chinese protocol of diagnosis and treatment of CRC (2020 edition) [2].

The HIPEC was implemented immediately after RS. For HIPEC, four catheters were placed in the abdominal cavity. Extracorporeal circulation pipes were connected to a HIPEC perfusion machine (BRTRG-II, Bright Medical Tech, Guangzhou, China) (Appendix A). The perfusion flow rate was 400 mL/min, the perfusion bag was filled with 2 L/m^2^ normal saline, the perfusion time was 60 min, and the perfusion temperature was 43 ± 0.5 ℃. HIPEC antitumor drugs chosen were mitomycin, lobaplatin, raltitrexed, or oxaliplatin, and the dosage was based on the patient’s body surface area (refer to guidelines for systemic chemotherapy [2,10]) (Appendix A).

Postoperative adjuvant chemotherapy started 3–4 weeks after HIPEC. Patients with poor physical condition could extend it appropriately, but it should not exceed 8 weeks after operation at the latest. The chemotherapy scheme was determined by clinicians according to the pathological stage, molecular classification, and risk factors, referring to NCCN Guidelines (2018) [10]. The XELOX or mFOLFOX6 scheme was preferred.

### 2.5. Statistical Analysis

The reporting standard of CTCAE 5.0 was used to define the HIPEC-related adverse events. OS and RFS were compared between RS and RS + HIEPC groups. A separate analysis was performed for tumor T-stage, tumor location, and tumor size. IBM SPSS Statistics version 24.0 (IBM Corp., Armonk, NY, USA) was used for statistical processing, and GraphPad Prism version 8.0 (GraphPad Software, San Diego, CA, USA) was used for mapping. Categorical data are expressed as percentages, continuous data as mean or median, normally distributed data as mean ± standard deviation (SD), and survival data as a Kaplan–Meier curve. Categorical variables were analyzed by means of the Pearson chi-square test or Fisher exact test, and continuous variables were analyzed by means of Student’s *t*-test or Mann–Whitney U test. The survival data were reported using Kaplan–Meier estimates, and differences were determined by the log-rank test. Propensity score matching (PSM) was used to reduce the influence of biases and confounding variables to make a more reasonable comparison between the experimental and control groups (Appendix A). A 1:1 PSM was conducted for 16 clinically relevant variables in Table 1. The matching quality was assessed using absolute standard differences, with a value < 5% considered insignificant. Statistical significance was set at two-sided *p*-values < 0.05.

## 3. Results

### 3.1. Baseline Clinicopathological Characteristics of Patients

The baseline clinicopathological characteristics of the patients are listed in Table 1. Using the inclusion and exclusion criteria, we identified 417 patients from 800, 165 managed with RS + HIPEC and 252 with RS alone (Figure 1). There were 263 men and 154 women, with a median age of 61 (24–86) years. After PSM, there were 246 patients in total, 150 men and 96 women, with a median age of 60 (24–86) years. Before PSM, there was no significant difference between the two groups with respect to sex (*p* = 0.208, χ^2^ = 1.584), CCI (*p* = 0.462, χ^2^ = 0.737), ASA (*p* = 0.494, χ^2^ = 0.685), tumor location (*p* = 0.247, χ^2^ = 2.800), tumor size (*p* = 0.177, Z = 1.353), tumor differentiation (*p* = 0.335, χ^2^ = 2.187), pN (*p* = 0.115, χ^2^ = 7.432), number of resected lymph nodes (*p* = 0.059, Z = 1.895), nerve invasion (*p* = 0.817, χ^2^ = 0.054), vascular invasion (*p* = 0.153, χ^2^ = 2.043), mismatch repair mutation (*p* = 0.897, χ^2^ = 0.017), and adjuvant chemotherapy (*p* = 0.256, χ^2^ = 2.722). However, age (*p* = 0.004, Z = −2.873), BMI (*p* = 0.007, Z = 2.725), surgical procedure (*p* < 0.001, χ^2^ = 20.609), pT (*p* < 0.001, χ^2^ = 20.531), and length of postoperative stay (*p* = 0.048, Z = −1.987) were significantly different between the RS + HIPEC and RS groups.

After PSM, there was no statistically significant difference in any of the baseline clinicopathological characteristics between the RS + HIPEC and RS groups. The baseline clinicopathological characteristics of the patients after PSM are listed in Table 2.

### 3.2. Adverse Events

The details and rates of adverse events globally or in the groups before and after PSM are listed in Table 3.

Before PSM, there was no significant difference between the RS + HIPEC and RS groups in the incidence of moderate or severe anemia (*p* = 0.968, χ^2^ = 0.002), hypoalbuminemia (*p* = 0.050, χ^2^ = 3.853), myelosuppression (*p* = 0.216, χ^2^ = 1.531), wound complications (*p* = 0.611, χ^2^ = 0.259), abdominal infection (*p* = 0.825, χ^2^ = 0.049), pulmonary infection (*p* = 0.764, χ^2^ = 0.090), postoperative bleeding (*p* = 0.668, χ^2^ = 0.184), anastomotic leakage (*p* = 0.753, χ^2^ = 0.009), ileus (*p* = 0.858, χ^2^ = 0.032), electrolyte disorder (*p* = 0.295, χ^2^ = 1.097), or abdominal discomfort (*p* = 0.321, χ^2^ = 0.983).

After PSM, there was no significant difference in the occurrence of any adverse events between the RS + HIPEC and RS groups.

### 3.3. Comparison of Prognosis

Follow-up data from 417 patients were included in the analysis. Before PSM, the RS + HIPEC group had a lower median follow-up time than the RS group, with 77.0 vs. 81.2% RFS and 91.9 vs. 92.3% OS at 24 months, in the RS + HIPEC and RS groups, respectively. The corresponding cumulative RFS and OS on the Kaplan–Meier survival curve were not significantly different (*p* = 0.643, χ^2^ = 0.215 and *p* = 0.066, χ^2^ = 3.381, respectively) (Figure 2).

After PSM, the RS + HIPEC group had a lower median follow-up time than the RS group, with 16 (12–27) and 21 (3–25) months, respectively. On the Kaplan–Meier survival curve, the RFS of both groups at 24 months (73.4 vs. 81.1%) showed no significant difference (*p* = 0.580, χ^2^ = 0.442). However, the OS of both groups at 24 months (82.0 vs. 88.5%) was not significantly different (*p* = 0.072, χ^2^ = 3.242) (Figure 2).

### 3.4. Stratified Analysis of Prognosis

Stratified analysis was used to further explore the factors influencing RFS and OS in patients with CRC. The patients were divided into T stage, tumor location, and size groups to determine the influence of these factors on effectiveness.

After PSM, when the patients had stage T4a tumors (*n* = 86), the RFS and OS of both groups (*n*RS + HIPEC = 46; *n*RS = 40) showed no significant difference (*p* = 0.652, χ^2^ = 0.203 and *p* = 0.282, χ^2^ = 1.156, respectively). For patients with stage T4b tumors (*n* = 160), the RS + HIPEC group (*n* = 77) had better cumulative RFS in 24 months (92.1 vs. 77.3%) than those in the RS group (*n* = 83) (*p* = 0.048, χ^2^ = 3.902) (Figure 3). However, the OS of both groups was not significantly different (*p* = 0.161, χ^2^ = 1.964) (Appendix A).

After PSM, for patients with colon cancer (*n* = 150), the RFS and OS of the two groups (*n*RS + HIPEC = 73; *n*RS = 77) showed no significant differences (*p* = 1.000, χ^2^ < 0.001 and *p* = 0.177, χ^2^ = 1.820, respectively). In addition, for patients with rectal cancer (*n* = 96), the RFS and OS of both groups (*n*RS + HIPEC = 50; *n*RS = 46) were not significantly different (*p* = 0.419, χ^2^ = 0.654 and *p* = 0.223, χ^2^ = 1.482, respectively) (Appendix A).

After PSM, with respect to the patients with tumor size < 5 cm (*n* = 142), the RFS and OS of both groups (*n*RS + HIPEC = 72; *n*RS = 70) showed no significant difference (*p* = 0.866, χ^2^ = 0.029 and *p* = 0.062, χ^2^ = 3.486, respectively). For patients with tumor size ≥ 5 cm (*n* = 104), the RS + HIPEC group (*n* = 51) had better cumulative RFS in 24 months (93.0 vs. 80.9%) than those in the RS group (*n* = 53) (*p* = 0.029, χ^2^ = 4.744) (Figure 4). However, the OS of both groups was not significantly different (*p* = 0.470, χ^2^ = 0.522) (Appendix A).

## 4. Discussion

Several reports have confirmed that HIPEC can clear free cancer cell in the peritoneal cavity and can be used as an effective adjuvant treatment for preventing PC after surgery for advanced peritoneal tumors [14,15,16,17,18]. Several clinical studies have reported that cytoreductive surgery combined with HIPEC could significantly improve OS or RFS in CRC with PC [19,20,21]. Since there is high-level clinical evidence supporting the therapeutic role of HIPEC in peritoneal metastasis of CRC, HIPEC has become the recommended method for CRC with PC in many countries [8,9,10]. Nevertheless, the use of HIPEC in T4 CRC is controversial. Our study confirmed the safety of RS combined with HIPEC in treating T4 CRC and its short-term efficacy in patients with T4b stage CRC and tumor size ≥ 5 cm.

Regarding the adverse events occurring in RS + HIPEC in patients with PC, Verwaal et al. reported grade III–IV adverse reactions, including leukopenia (17%), gastrointestinal fistula (15%), hemorrhage (14%), and heart failure (12%) [20]. Goere et al. reported that grade III–IV postoperative complications occurred in 41% of patients who underwent second-look surgery and HIPEC [22]. Quenet et al. reported that at 60 days postoperatively, the grade III–V morbidity rate in the comparator arm was significantly higher than that in the HIPEC arm (24.1 vs. 13.6%, *p* = 0.030) [19]. Klaver et al. reported that HIPEC-related complications occurred in 14% of patients who underwent HIPEC [11]. HIPEC increased the risk of complications in patients with CRC, which was different from that in the control group, but it was within the acceptable range. In our study, the two most common complications were hypoalbuminemia (19.5%) and anemia (17.9%) in the RS + HIPEC group. After PSM, there was no significant difference in the statistics for all adverse events between two groups. Among these adverse events, the most concerning to clinicians is the occurrence of gastrointestinal anastomotic leakage after HIPEC treatment. Similarly, recent research has indicated that the occurrence of gastrointestinal anastomotic leakage has no clear relationship with HIPEC treatment. Additionally, HIPEC does not increase the risk of anastomotic leakage compared to conventional gastrointestinal surgery [7,11,23,24]. It is feasible and stable to carry out HIPEC by experienced medical and nursing staff following technical standards.

Since HIPEC was applied in patients with CRC, much research has focused on the effectiveness of HIPEC in patients with CRC. Our study showed that the RFS (73.4 vs. 81.1%, *p* = 0.580) and OS (82.0 vs. 88.5%, *p* = 0.072) between RS + HIPEC and RS groups on overall analysis did not show significant differences before and after PSM. The COLOPEC trial showed no difference in disease-free survival (DFS) at 18 months (69.0 vs. 69.3%; *p* = 0.99), OS at 18 months (93.0 vs. 94.1%; *p* = 0.82), or peritoneal metastases-free survival at 18 months (80.9 vs. 76.2%, *p* = 0.28) between the experimental and control groups [11]. The PROPHYLOCHIP trial showed no difference in the 5-year DFS (42 vs. 49%, *p* = 0.82) and 5-year OS (68 vs. 72%, *p*-value not reported) between the HIPEC and control groups after a median follow-up of 50.8 months [22,25]. Both RCTs investigated the preventive effect of HIPEC in patients with CRC with high risk factors for PC. The effectiveness of HIPEC in preventing PC in CRC has not been presented in terms of currently reported results. Nonetheless, results of HIPECT4 showed that RS + HIEPC for locally advanced colon cancer could improve the loco-regional control rate (35.3 ± 0.4 vs. 33.2 ± 0.8 months), with 3-year loco-regional control rates of 97% and 87% (*p* = 0.025) [12]. Meanwhile, our research showed that HIPEC could improve the RFS of T4 CRC patients with T4b stage CRC (92.1 vs. 77.3%, *p* = 0.048) and tumor size ≥ 5 cm at 24 months (93.0 vs. 80.9%, *p* = 0.029).

Patient selection is an important factor that influences the efficacy of HIPEC in preventing PC. For patients with T4 stage CRC, D. Hompes et al. showed that the incidence of PC was 13.2% at 3 years after RS in stage II-III CRC, including 20–36.7% in pT4 stage [6]. Even after RS and regular adjuvant chemotherapy, the 3-year DFS of patients with pT4 CRC is only 58–61%, which is far lower than that of patients with T1, T2, or T3 CRC [26]. A retrospective control study showed that among patients with T4 CRC with high risk factors of peritoneal metastasis (T3/4, mucinous adenocarcinoma/signet ring cell carcinoma), the risk of peritoneal metastasis in the treatment group receiving RS combined with HIPEC was 4%, while the risk in the control group without HIPEC was 28% [27]. It suggested that the use of HIPEC in selective population may help to reduce the risk of PC. Previous studies have shown that the T stage is an important factor that affects the prognosis of patients with locally advanced gastric cancer after surgery with prophylactic HIPEC [28]. However, few studies have reported the effect of T staging and tumor size on the application of HIPEC in T4 CRC. After a comprehensive follow-up, we confirmed the role of HIPEC in patients with T4b stage CRC and tumor size ≥ 5 cm. These patient characteristics can be a predictor of HIPEC effectiveness and define the possible window of opportunity for HIPEC to prevent colorectal PC.

This study had some limitations. First, it was retrospective, and its conclusions need to be verified by prospective cohort studies. Second, the follow-up period was not long enough to obtain a long-term prognosis. Finally, the shorter follow-up period may overstate the short-term prognosis, requiring further long-term follow-up and more subjects with specific stratification in the later stage.

## 5. Conclusions

Thus, the safety of HIPEC in patients with T4 CRC can be guaranteed. Furthermore, regarding prognosis, RS + HIPEC did not show the expected advantages in the overall cohort at 24 months, and a longer-term follow-up is required. However, patients with T4 CRC who had T4b stage CRC and tumor size ≥ 5 cm and underwent RS combined with HIPEC had better RFS than patients who underwent simple RS at 24 months.

## Figures and Tables

**Figure 1 jcm-12-01145-f001:**
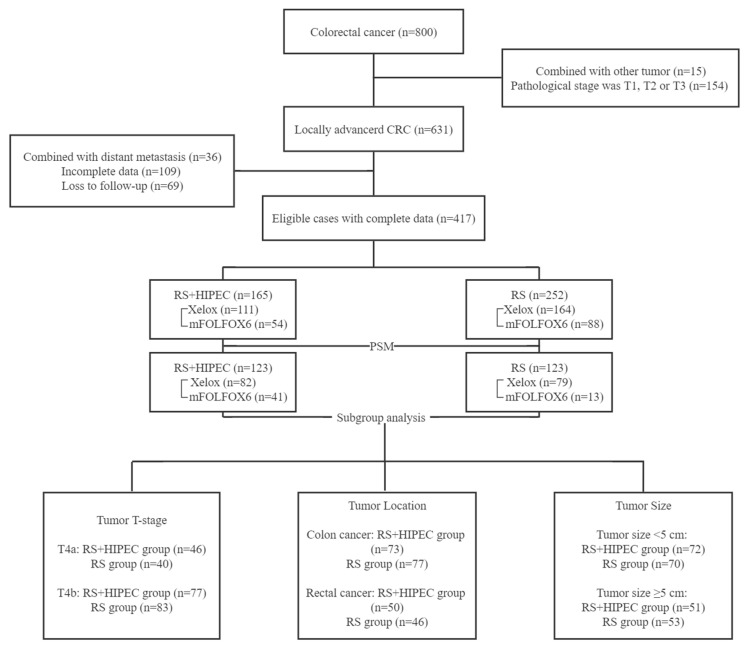
Flow chart. A total of 417 out of 800 T4 CRC patients were included in the study. Inclusion criteria: (1) confirmation of colorectal adenocarcinoma by postoperative pathology (pT ≥ 4), (2) no distant metastasis or PC detected before the operation through computed tomography (CT) or magnetic resonance imaging (MRI), (3) adjuvant chemotherapy taken after discharge, and (4) a tumor location in patients with rectal cancer above the peritoneal reflection. Preoperative MRI or colonoscopy showed at least 7 cm from the lower border of the tumor to the anal verge (above the peritoneal reflection). The exclusion criteria included: (1) incomplete clinical and pathological data preoperatively, (2) tumor metastasis found during the operation, (3) patients with a history of other malignancies within 5 years, and (4) emergency surgery due to perforation, ileus, etc.

**Figure 2 jcm-12-01145-f002:**
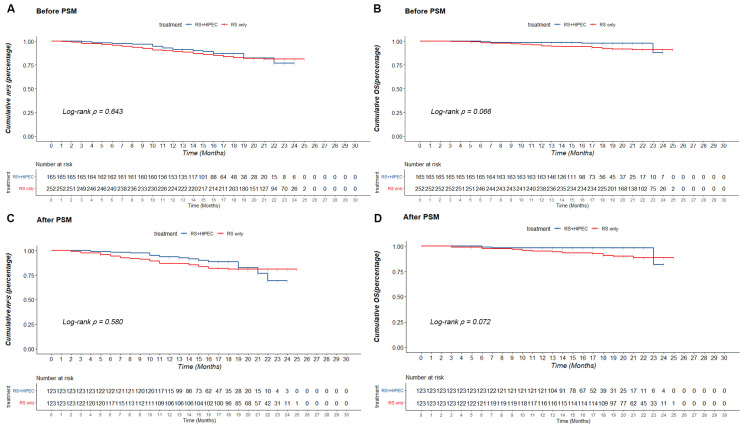
Kaplan–Meier method was used to compare recurrence-free survival (RFS) and overall survival (OS) between RS + HIPEC group and RS group. (**A**) RFS of RS + HIPEC group and RS group before PSM. (**B**) OS of RS + HIPEC group and RS group before PSM. (**C**) RFS of RS + HIPEC group and RS group after PSM. (**D**) OS of RS + HIPEC group and RS group after PSM.

**Figure 3 jcm-12-01145-f003:**
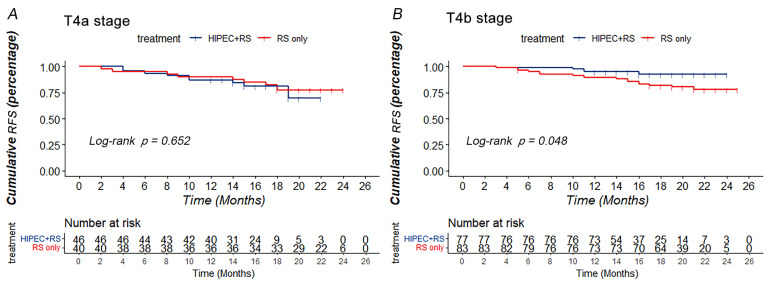
Subgroup analysis based on T stage. (**A**) RFS of RS + HIPEC group and RS group after PSM when the tumor was at T4a stage. (**B**) RFS of RS + HIPEC group and RS group after PSM when the tumor was at T4b stage.

**Figure 4 jcm-12-01145-f004:**
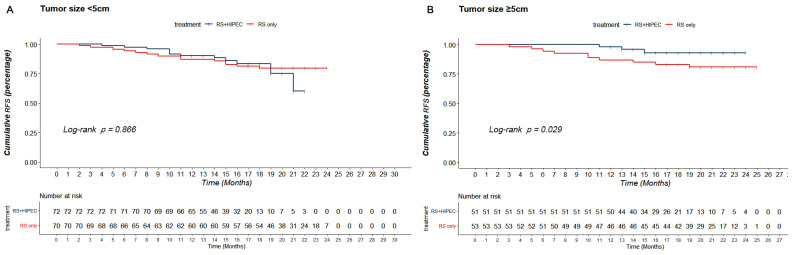
Subgroup analysis based on tumor size. (**A**) RFS of RS + HIPEC group and RS group after PSM when the tumor was <5 cm. (**B**) RFS of RS + HIPEC group and RS group after PSM when the tumor was ≥5 cm.

**Table 1 jcm-12-01145-t001:** Baseline clinicopathological characteristics of patients with T4 colorectal cancer before PSM.

Characteristic	No (%)			χ^2^/Z	*p*-Value
	Overall (*n* = 417)	RS + HIPEC (*n* = 165)	RS Alone (*n* = 252)		
Age, years					
Mean ± SD	59.78 ± 12.319	57.66 ± 12.301	61.17 ± 12.156	−2.873	0.004
≤60	204 (48.9)	86 (52.1)	118 (46.8)		
>60	213 (51.1)	79 (47.9)	134 (53.2)		
Sex				1.584	0.208
Female	154 (36.9)	67 (40.6)	87 (34.5)		
Male	263 (63.1)	98 (59.4)	165 (65.5)		
BMI					
Mean ± SD	22.74 ± 3.492	23.31 ± 3.318	22.37 ± 3.559	2.725	0.007
Charlson Comorbidity Index				0.737	0.462
0	307 (73.6)	119 (72.1)	188 (74.6)		
1	77 (18.5)	32 (19.4)	45 (17.9)		
2	21 (5.0)	8 (4.8)	13 (5.2)		
≥3	12 (2.9)	6 (3.6)	6 (2.4)		
ASA Score				0.685	0.494
1	89 (21.3)	29 (17.6)	60 (23.8)		
2	303 (72.7)	128 (77.6)	175 (69.4)		
≥3	25 (6.0)	8 (4.8)	17 (6.8)		
Surgical Procedures				20.609	<0.001
laparoscopy	371 (89.0)	161 (97.6)	210 (83.3)		
laparotomy	46 (11.0)	4 (2.4)	42 (16.7)		
Tumor Location				2.800	0.247
right semicolon	105 (25.2)	45 (27.3)	60 (23.8)		
left semicolon	137 (32.9)	59 (35.8)	78 (31.0)		
rectum	175 (42.0)	61 (37.0)	114 (45.2)		
Tumor Size, cm					
Mean ± SD	4.764 ± 2.219	4.945 ± 2.510	4.645 ± 2.002	1.353	0.177
<5	248 (59.5)	93 (56.4)	155 (61.5)		
≥5	169 (40.5)	72 (43.6)	97 (38.5)		
Tumor differentiation				2.187	0.335
poor or undifferentiation	29 (7.0)	11 (6.7)	18 (7.1)		
Well or moderately	388 (93.0)	154 (93.3)	234 (92.9)		
pT status				17.490	<0.001
pT4a	217 (52.0)	65 (39.4)	152 (60.3)		
pT4b	200 (48.0)	100 (60.6)	100 (39.7)		
No. of resected lymph nodes				1.895	0.059
Mean ± SD	20.32 ± 8.853	21.33 ± 7.493	19.66 ± 9.597		
pN status				7.432	0.115
pN0	234 (56.1)	89 (53.9)	145 (57.5)		
pN1a	50 (12.0)	19 (11.5)	31 (12.3)		
pN1b	58 (13.9)	32 (19.4)	26 (10.3)		
pN2a	38 (9.1)	13 (7.9)	25 (9.9)		
pN2b	37 (8.9)	12 (7.3)	25 (9.9)		
nerve invasion				0.054	0.817
No	260 (62.4)	104 (63.0)	156 (61.9)		
Yes	157 (37.6)	61 (37.0)	96 (38.1)		
vascular invasion				2.043	0.153
No	292 (70.0)	109 (66.1)	183 (72.6)		
Yes	125 (30.0)	56 (33.9)	69 (27.4)		
MMR positive				0.017	0.897
No	390 (93.5)	154 (93.3)	236 (93.7)		
Yes	27 (6.5)	11 (6.7)	16 (6.3)		
Post-surgery stay time					
Mean ± SD	12.20 ± 6.769	11.39 ± 3.852	12.73 ± 8.095	−1.987	0.048
Adjuvant chemotherapy				2.722	0.256
XELOX	275 (65.9)	111 (67.3)	164 (65.1)		
mFOLFOX6	142 (34.1)	54 (32.7)	88 (34.9)		
Follow-up time, months					
Mean ± SD	18.95 ± 3.857	16.66 ± 3.449	20.45 ± 3.345		
Median (range)	20 (3–27)	16 (12–27)	21 (3–25)		

PSM: propensity score matching; SD: standard deviation; BMI: body mass index; ASA: American Society of Anesthesiologists; MMR: mismatch repair mutation.

**Table 2 jcm-12-01145-t002:** Baseline clinicopathological characteristics of patients with T4 colorectal cancer after PSM.

Characteristic	No (%)			χ^2^/Z	*p*-Value
	Overall (*n* = 246)	RS + HIPEC (*n* = 123)	RS Alone (*n* = 123)		
Age, years					
Mean ± SD	59.03 ± 12.238	58.49 ± 11.554	59.58 ± 12.910	−0.697	0.486
≤60	125 (50.8)	60 (48.8)	65 (52.8)		
>60	121 (49.2)	63 (51.2)	58 (47.2)		
Sex				0.273	0.601
Female	96 (39.0)	50 (40.7)	46 (37.4)		
Male	150 (61.0)	73 (59.3)	77 (62.6)		
BMI					
Mean ± SD	22.875 ± 3.434	22.839 ± 3.158	22.912 ± 3.702	−0.168	0.867
Charlson Comorbidity Index				1.287	0.864
0	182 (74.0)	91 (74.0)	91 (74.0)		
1	46 (18.7)	22 (17.9)	24 (19.5)		
2	12 (4.9)	6 (4.9)	6 (4.9)		
≥3	6 (2.4)	4 (3.2)	2 (1.6)		
ASA Score				1.092	0.779
1	46 (18.7)	24 (19.5)	22 (17.9)		
2	185 (75.2)	92 (74.8)	93 (75.6)		
3	15 (6.1)	7 (5.7)	8 (6.5)		
Surgical Procedures				0.147	0.701
laparoscopy	239 (97.2)	119 (96.7)	120 (97.6)		
laparotomy	7 (2.8)	4 (3.3)	3 (2.4)		
Tumor Location				0.367	0.836
right semicolon	66 (26.8)	33 (26.8)	33 (26.8)		
left semicolon	84 (34.1)	40 (32.5)	44 (35.8)		
rectum	96 (39.0)	50 (40.7)	46 (37.4)		
Tumor Size, cm					
Mean ± SD	4.797 ± 2.349	4.74 ± 2.393	4.85 ± 2.313	−0.366	0.715
<5	142 (57.7)	72 (58.5)	70 (56.9)		
≥5	104 (42.3)	51 (41.5)	53 (43.1)		
Tumor differentiation				0.119	0.942
poor or undifferentiation	19 (7.7)	10 (8.1)	9 (7.3)		
Well or moderately	227 (92.3)	113 (91.9)	114 (92.7)		
pT status				0.644	0.422
pT4a	86 (35.0)	46 (37.4)	40 (32.5)		
pT4b	160 (65.0)	77 (62.6)	83 (67.5)		
No. of resected lymph nodes				−0.172	0.863
Mean ± SD	21.08 ± 9.969	20.97 ± 7.148	21.19 ± 12.185		
pN status				1.976	0.740
pN0	138 (56.1)	66 (53.7)	72 (58.5)		
pN1a	31 (12.6)	14 (11.4)	17 (13.8)		
pN1b	37 (15.0)	22 (17.9)	15 (12.2)		
pN2a	21 (8.5)	11 (8.9)	10 (8.1)		
pN2b	19 (7.7)	10 (8.1)	9 (7.3)		
nerve invasion				1.829	0.176
No	164 (66.7)	77 (62.6)	87 (70.7)		
Yes	82 (33.3)	46 (37.4)	36 (29.3)		
vascular invasion				0.072	0.788
No	162 (65.9)	80 (65.0)	82 (66.7)		
Yes	84 (34.1)	43 (35.0)	41 (33.3)		
MMR positive				0.000	1.000
No	230 (93.5)	115 (93.5)	115 (93.5)		
Yes	16 (6.5)	8 (6.5)	8 (6.5)		
Post-surgery stay time					
Mean ± SD	11.75 ± 7.842	10.84 ± 3.379	12.66 ± 10.507	−1.830	0.068
Adjuvant chemotherapy				2.884	0.236
XELOX	161 (65.4)	82 (66.7)	79 (64.2)		
mFOLFOX6	85 (34.6)	41 (33.3)	44 (35.8)		
Follow-up time, months					
Mean ± SD	18.32 ± 3.925	16.37 ± 3.431	20.28 ± 3.383		
Median (range)	19 (3–27)	16 (12–27)	21 (3–25)		

PSM: propensity score matching; SD: standard deviation; BMI: body mass index; ASA: American Society of Anesthesiologists; MMR: mismatch repair mutation.

**Table 3 jcm-12-01145-t003:** Adverse events of patients with T4 colorectal cancer before and after PSM.

Adverse Event	Before PSM					After PSM				
	No (%)			χ^2^	*p*-Value	No (%)			χ^2^	*p*-Value
	Overall (*n* = 417)	RS + HIPEC (*n* = 165)	RS Alone (*n* = 232)			Overall (*n* = 246)	RS + HIPEC (*n* = 123)	RS Alone (*n* = 123)		
Anemia	83 (19.9)	33 (20.0)	50 (19.8)	0.002	0.968	46 (18.7)	22 (17.9)	24 (19.5)	0.107	0.744
Hypoalbuminemia	113 (27.1)	36 (21.8)	77 (30.6)	3.853	0.050	59 (24.0)	24 (19.5)	35 (28.5)	2.698	0.100
Myelosuppression	1 (0.2)	1 (0.6)	0 (0.0)	1.531	0.216	1 (0.4)	1 (0.8)	0 (0.0)	1.004	0.316
Wound complications	20 (4.8)	9 (5.5)	11 (4.4)	0.259	0.611	9 (3.7)	4 (3.3)	5 (4.1)	0.115	0.734
Abdomen infection	3 (0.7)	1 (0.6)	2 (0.8)	0.049	0.825	1 (0.4)	0 (0.0)	1 (0.8)	1.004	0.316
Pulmonary infection	14 (3.4)	5 (3.0)	9 (3.6)	0.090	0.764	11 (4.5)	5 (4.1)	6 (4.9)	0.095	0.758
Postoperative bleeding	4 (1.0)	2 (1.2)	2 (0.8)	0.184	0.668	2 (0.8)	1 (0.8)	1 (0.8)	0.000	1.000
Anastomotic leakage	6 (1.4)	2 (1.2)	4 (1.6)	0.009	0.753	4 (1.6)	1 (0.8)	3 (2.4)	1.017	0.313
Ileus	7 (1.7)	3 (1.8)	4 (1.6)	0.032	0.858	7 (2.8)	3 (2.4)	4 (3.3)	0.147	0.701
Electrolyte disturbance	12 (2.9)	3 (1.8)	9 (3.6)	1.097	0.295	6 (2.4)	2 (1.6)	4 (3.3)	0.683	0.408
Abdomen discomfort	9 (2.2)	5 (3.0)	4 (1.6)	0.983	0.321	5 (2.0)	2 (1.6)	3 (2.4)	0.204	0.651

PSM: propensity score matching.

## Data Availability

The original anonymous dataset is available on request from the corresponding author.

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
