# Peer review of "Comparative Study of Short-Term Efficacy and Safety of Radical Surgery with or without Hyperthermic Intraperitoneal Chemotherapy in Colorectal Cancer with T4 Stage: A Propensity Score Matching Analysis"

_jcm, 2023, doi:10.3390/jcm12031145_

Round 1
Reviewer 1 Report
This manuscript describes a series of patients operated for stage II, T4 colorectal cancer in a high-volume unit. Some on the patients received HIPEC while others did not. The main message of this study is the find out whether prophylactic HIPEC is useful for T4 patients. The authors discuss the existing literature with some conflicting results and can well justify the importance of this work. I congratulate the authors for analysing this material and writing the manuscript.
I personally see two major shortcomings in this manuscript. First, the reason for selecting which patient underwent HIPEC is not clear. It may be difficult to find out the exact reasons for every single patient especially in a retrospective study, but as this is a single centre study and the authors represent the study site, they should be able to provide a more detailed discussion on patient selection for prophylactic HIPEC. Second, as the authors admit, the follow-up time is all too short to make any definite conclusions, but this will hopefully be addressed in a future study with longer follow-up. I would like to see the 5-year results of these patients.
Other concerns:
Introduction
“The latest data shows the United States had the third highest incidence of CRC in 2021 in both males and females(1).” This meaning does not really make sense. Should it be something like “…CRC had the third highest incidence among all cancers in both males and females.”?
Study cohort
It should be clearly stated here that the study patients did not have detectable carcinomatosis. Also, if I understand right, the authors retrospectively excluded all the patients with T1-T3 disease. This should be discussed, because the exact T class is not always evident preoperatively and this data cannot be used in clinical decision making without knowing the T class.
“All study participants provided full informed consent.” The Ethics chapter in the end says that written informed consents were not needed. Please clarify this.
Treatment
Were the primary tumours always resected in the same operation in which HIPEC was done? Was any peritonectomy done? What was the reason for choosing to do or not to do HIPEC? According to the tables most of the operations were laparoscopic all the way through including the HIPEC. Was that so? Could well be, but as peritonectomy with HIPEC is often done as open operation, this difference should be highlighted.
Propensity score matching
I read this manuscript from clinician’s, not statistician’s, point of view, but I like the use of double statistics with both partial propensity score matching and unmatched cohorts. The propensity score matched groups seem to be pretty similar, but a large number of patients is left out, which could confound the results.
Results
Short follow-up times. Does the perceived RFS difference result in OS difference later on?
Discussion
Generally, I think the discussion fits the purpose.
“Similarly, recent researches have indicated that the occurrence of gastrointestinal anastomotic leakage has no clear relationship with HIPEC treatment.” This sentence should have a reference of its own.
“…the loco-regional control rate (35,3±0.4 vs. 33.2±0.8 months), with 3-year locoregional control rates of 97% and 87% (p= 0.025) (12).” What do those months mean?
“…is only 58-61%, which is far lower than that of patients with other stages(26).” The use of word “stage” is misleading here.
Overall
English language editing is needed and there are a few sentences where the reader will have to guess what the authors probably are trying to say. However, the text is quite readable, and the message is clear.
Author Response
Dear Reviewer,
On behalf of my co-authors, we would like to thank you for taking the time to review our manuscript. Thanks for your comments concerning our manuscript entitled “Comparative Study of Short-term Efficacy and Safety of Radical Surgery with or without Hyperthermic Intraperitoneal Chemotherapy in Colorectal Cancer with T4 stage: A Propensity Score Matching Analysis” (Manuscript ID: jcm-2134670). Those comments are all valuable and helpful for revising and improving our paper. We have studied all comments carefully and have made conscientious correction. Revised portion are highlighted by using the track changes mode. The comments are reproduced and our responses are given directly afterward in red color.
Point 1: First, the reason for selecting which patient underwent HIPEC is not clear. It may be difficult to find out the exact reasons for every single patient especially in a retrospective study, but as this is a single centre study and the authors represent the study site, they should be able to provide a more detailed discussion on patient selection for prophylactic HIPEC.
Response 1: We appreciate you for this helpful recommendation and valuable suggestion. Our study aimed to describe the short-term efficacy and safety of prophylactic HIPEC after RS in patients with T4 CRC. So, we select pT4 patients in our research. In the center, if patients were diagnosed with cT4 colorectal cancer before surgery, we would evaluate their general condition and assess whether or not to recommend HIPEC. The treatment strategy was discussed preoperatively together with patients and their relatives. HIPEC was implemented after patients signed informed consent (line 128-130). Not all patients with T4 colorectal cancer would choose to receive HIPEC. The survival and recurrence of T4 patients who did not receive HIPEC is always our concern. HIPEC-06 prospective RCT (NCT 04370925) “A Multicenter Prospective Randomized Controlled Clinical Trial of Hyperthermic Intraperitoneal Chemotherapy after Colectomy in Patients with Colorectal Cancer at High Risk of Peritoneal Carcinomatosis” participated by our center is studying this topic.
Point 2: Second, as the authors admit, the follow-up time is all too short to make any definite conclusions, but this will hopefully be addressed in a future study with longer follow-up. I would like to see the 5-year results of these patients.
Response 2: Thank you for this valuable feedback and we believe that your comments shall be very useful to our further exploration in future. We certainly will conduct long-term follow-up on these patients.
Introduction
Point 3: “The latest data shows the United States had the third highest incidence of CRC in 2021 in both males and females(1).” This meaning does not really make sense. Should it be something like “…CRC had the third highest incidence among all cancers in both males and females.”?
Response 3: We have made the change according to the comment.
Study cohort
Point 4: It should be clearly stated here that the study patients did not have detectable carcinomatosis. Also, if I understand right, the authors retrospectively excluded all the patients with T1-T3 disease. This should be discussed, because the exact T class is not always evident preoperatively and this data cannot be used in clinical decision making without knowing the T class.
Response 4: We thank you for pointing this out. All the patients included did not have detectable carcinomatosis, we check the information before and after surgery (line 79-81, 85-86) “no distant metastasis or PC detected before the operation through computed tomography (CT) or magnetic resonance imaging (MRI)” “tumor metastasis found during the operation”. And we made the change in exclusion. In addition, as we responded in point 1, if patients were diagnosed with cT4 colorectal cancer before surgery, we would evaluate their general condition and assess whether or not to recommend HIPEC. We would not recommend HIPEC to patients if cT1\2\3 was diagnosed preoperatively. Because the pathological report of the pathology department takes 2-3 weeks or even longer, we will not implement HIPEC until the pathological report comes out. We have registered a prospective RCT (NCT04845490), including all cT4 patients received HIPEC, in order to further confirm the results of this study. We expect HIEPC to benefit patients with cT4, including those with unconfirmed pT4.
Point 5: “All study participants provided full informed consent.” The Ethics chapter in the end says that written informed consents were not needed. Please clarify this.
Response 5: We are very sorry for our clerical error. We have made the correction. Informed consent provided for surgery and HIPEC not for the study (line 127-129). And the data were collected anonymized and centrally, and the requirement for written informed consent was therefore waived.
Treatment
Point 6: Were the primary tumours always resected in the same operation in which HIPEC was done? Was any peritonectomy done? What was the reason for choosing to do or not to do HIPEC? According to the tables most of the operations were laparoscopic all the way through including the HIPEC. Was that so? Could well be, but as peritonectomy with HIPEC is often done as open operation, this difference should be highlighted.
Response 6: We thank you for raising this question. The patients enrolled in the study received radical surgery instead of cytoreductive surgery, surgery procedure as we mentioned “Clinicians performed radical resection of CRC (resection of the corresponding colon and rectum plus regional lymph node dissection), following the principles of mesorectal excision and tumour free operation, and the surgical procedure was referred to Chinese protocol of diagnosis and treatment of CRC (2020 edition)”. As we showed in Supplementary Figure 1, we performed HIPEC in closed way.
Clinically, we discuss the implementation of HIPEC with patients diagnosed with cT4 preoperatively by imaging and endoscopy. HIPEC was implemented after patients signed informed consent. We would not recommend HIPEC to patients if cT3 was diagnosed preoperatively. Because the pathological report of the pathology department takes 2-3 weeks or even longer, we will not implement HIPEC until the pathological report comes out. As you comment, how to accurately determine T4 tumors before pathological samples are obtained is also a challenge to clinician. This resulted in patients with a clinical diagnosis of T4, in fact with a pathological diagnosis of T3, receiving HIPEC.
Propensity score matching
Point 7: I read this manuscript from clinician’s, not statistician’s, point of view, but I like the use of double statistics with both partial propensity score matching and unmatched cohorts. The propensity score matched groups seem to be pretty similar, but a large number of patients is left out, which could confound the results.
Response 7: We thank you for pointing this out. As you comment, we analyzed OS and RFS of overall cohort before PSM. And PSM is used to process subgroup data. In the observation study, due to various reasons, there are many bias and confounding variables. The method of PSM is to reduce the impact of these bias and confounding variables, so as to make a more reasonable comparison between the groups.
Results
Point 8: Short follow-up times. Does the perceived RFS difference result in OS difference later on?
Response 8: Thank you for this insightful question. Our study indeed had this limitation, we have discussed short-term follow-up in manuscript: “Second, the follow-up period was not long enough to obtain a long-term prognosis. Finally, the shorter follow-up period may overstate the short-term prognosis, requiring further long-term follow-up and more subjects with specific stratification in the later stage” (line 315-318) In future, we certainly will conduct long-term follow-up on the subjects and carry out relevant prospective clinical studies to further explore the application of drugs in HIEPC (NCT04845490, NCT04808466).
Discussion
Point 9: “Similarly, recent researches have indicated that the occurrence of gastrointestinal anastomotic leakage has no clear relationship with HIPEC treatment.” This sentence should have a reference of its own.
Response 9: Thanks for comment. The reference had been attached in the following sentence “Similarly, recent researches have indicated that the occurrence of gastrointestinal anastomotic leakage has no clear relationship with HIPEC treatment. Additionally, the HIPEC does not increase the risk of anastomotic leakage compared to conventional gastrointestinal surgery. (7, 11, 23, 24).”
Point 10: “…the loco-regional control rate (35,3±0.4 vs. 33.2±0.8 months), with 3-year locoregional control rates of 97% and 87% (p= 0.025) (12).” What do those months mean?
Response 10: The original text is as follows “The loco-regional control (LC) was improved in the experimental arm (35.3 ± 0.4 vs. 33.2 ± 0.8 months) with a 3 years LC rate of 97% vs. 87% (p ¼ 0.025). No differences were observed in DFS and OS. The pT4 subgroup showed a clear benefit of LC in the HIPEC arm. No differences in morbidity were observed between groups.” It may mean the recurrence-free survival time.
Point 11: “…is only 58-61%, which is far lower than that of patients with other stages(26).” The use of word “stage” is misleading here.
Response 11: We have made the change “which is far lower than that of patients with T1, T2 or T3 CRC”.
We would like to express our great appreciation to you for comments on our paper. Looking forward to meet your approval.
Thank you and best regards.

Reviewer 2 Report
Although the present article contains valuable information, several issues were encountered, which raise serious concerns regarding the scientific validity of the presented information:
1. The primary endpoint of overall survival at 24 months is impossible to achieve for a group of patients operated between January 2020 and April 2021.
2. The patient inclusion and exclusion criteria are not adequately explained.
3. The figures are not appropriately mentioned in the text and are lacking adequate descriptions.
4. The statistical analysis is not adequately described.
5. Lastly, while the use of language is mostly sound, there are several instances where certain points are unclear, making the narrative difficult to follow. A revision of grammar and syntax is required in order to improve the flow and readability of the text.
Author Response
Dear reviewer,
On behalf of my co-authors, we thank you very much for giving us an opportunity to revise our manuscript, we appreciate you very much for positive and constructive comments and suggestions on our manuscript entitled “Comparative Study of Short-term Efficacy and Safety of Radical Surgery with or without Hyperthermic Intraperitoneal Chemotherapy in Colorectal Cancer with T4 stage: A Propensity Score Matching Analysis” (Manuscript ID: jcm-2134670). Your invaluable and useful suggestions have enabled us to improve our work. Based on the instructions provided in your comments, we uploaded the file of the original manuscript with all the changes highlighted by using the track changes mode. The comments are reproduced and our responses are given directly afterward in red. Thanks again!
Point 1: The primary endpoint of overall survival at 24 months is impossible to achieve for a group of patients operated between January 2020 and April 2021.
Response 1: We really appreciate you for this comment. It was our mistake to write the wrong year. Actually, we reviewed database from January 2019 to April 2020. Owing to the outbreak and spread of COVID-19 in Wuhan, the number of patients coming to the hospital has decreased a lot since April 2020. So, data collection is up to April 2020.
Point 2: The patient inclusion and exclusion criteria are not adequately explained.
Response 2: We apologized for this. Could you give us more detailed suggestions on which criteria confuse you?
Point 3: The figures are not appropriately mentioned in the text and are lacking adequate descriptions.
Response 3: Thanks for the comment. Would you please point out which kind of information need to be added in the article? We had written figure legends after Reference to describe the figures.
Point 4: The statistical analysis is not adequately described.
Response 4: Thank you for this valuable comment. We have rewritten the part of “Statistical analysis”.
We would like to express our great appreciation to you for comments on our paper. Looking forward to meet your approval.
Thank you and best regards.

Round 2
Reviewer 2 Report
the manuscript has been sufficiently improved